# Comfortably Numb? Researchers' Satisfaction with the Publication System and a Proposal for Radical Change

**Hans van Dijk** *  **and Marino van Zelst**

Tilburg University, Department of Organization Studies, 5037 AB Tilburg, The Netherlands; j.m.vanzelst@uvt.nl
* Correspondence: j.vandijk1@uvt.nl

**Abstract:** In this preregistered study we evaluate current attitudes towards, and experiences with, publishing research and propose an alternative system of publishing. Our main hypothesis is that researchers tend to become institutionalized, such that they are generally discontent with the current publication system, but that this dissatisfaction fades over time as they become tenured. A survey was distributed to the first authors of papers published in four recent issues of top-15 Work and Organizational Psychology (WOP) journals. Even among this positively biased sample, we found that the time it takes to publish a manuscript is negatively associated with whether authors perceive this time to be justifiable and worthwhile relative to the amount their manuscript has changed. Review quality and tenure buffer the negative relationship with perceived justifiability, but not for perceived worth. The findings suggest that untenured (WOP) researchers are dissatisfied with the publishing times of academic journals, which adds to the pile of criticisms of the journal-based publication system. Since publishing times are inherent to the journal-based publication system, we suggest that incremental improvements may not sufficiently address the problems associated with publishing times. We therefore propose the adoption of a modular publication system to improve (WOP) publishing experiences.

**Keywords:** meta–science; publication system; publishing times; satisfaction

---

> When I was a child I caught a fleeting glimpse,
>
> Out of the corner of my eye.
>
> I turned to look but it was gone.
>
> I cannot put my finger on it now.
>
> The child is grown, the dream is gone.
>
> I have become comfortably numb.
>
> *Pink Floyd—Comfortably Numb*

## 1. Introduction

In 2015 the hashtag #IAmAScientistBecause generated a massive response [1]. Researchers across the globe shared the motivation behind their vocational choice. Two often heard responses revolved around the curiosity to understand and explain what is yet unknown, and to find solutions to problems that are yet unsolved. Such intrinsic motivations are known to be predictive of high levels of job satisfaction, commitment, and health [2].

However, there are also numerous accounts of academics who gave up their jobs. Often-mentioned reasons are hypercompetition, perverse incentives in academia, and incremental science [3–5]—factors

that are known to hamper intrinsic motivation [6]. In this paper, we argue that these and related demotivating aspects of academia find their origin in the journal-based publication system. Since researchers are evaluated mainly based on where and how much they publish, the publication system is of primary importance to researchers' job motivation and satisfaction [7].

In the last decades, this publication system has come under strain: English has become the lingua franca of scholarly communication, causing an increasing number of institutions across the globe to prioritize publications in English-language journals over publications in other languages. The corresponding increased scarcity of spots for publications in combination with the globalization of the academic job market has led to (a sense of) hypercompetition among academics [3]. In combination with several other, often mentioned design flaws in the publication system (e.g., long publishing times, flawed quality indicators based on which individual researchers are evaluated, biased reviews, see e.g., [8]), it stands to reason that the publication system is a prominent source of dissatisfaction among researchers. However, there is scant empirical research that has examined researchers' satisfaction with the publication system. Our first aim is therefore to fill that gap and delve deeper into researchers' perceptions of the publishing process.

Specifically, in this study we focus on publishing time (defined as the time between paper submission and acceptance for publication) as a major source of dissatisfaction with the publication system. Publishing time is highly consequential for the careers of researchers (e.g., getting a timely response can mean the difference between getting tenure or not), but out of their own control, which makes it a prominent source of stress and dissatisfaction [9]. Based on justice theory, which suggests that procedural and distributive justice shape outcome satisfaction [10], we specifically argue that publishing time negatively affects the extent to which researchers perceive publishing times to be worthwhile (as indicative of procedural justice, defined as the perceived fairness of outcomes [11]) and justifiable (as indicative of procedural justice, i.e., the perceived fairness of the means used to determine outcomes [10]). We also expect that these relationships are buffered by the quality of the reviews and the impact factor of the journal[1] in which a paper is published, respectively.

Based on insights from system justification theory [12], we further hypothesize that researchers without tenure, in particular, are likely to become more dissatisfied when publishing time takes longer, in part because the outcomes are more important to the careers of researchers without tenure, and in part because researchers with tenure may have grown accustomed to long publishing times and derive their academic standing from the current publication system. We argue that the difference in experiences between tenured and untenured academics is of pivotal importance, given that it suggests that those in charge of the publication system (e.g., editors) as well as those who trust in the publication system for evaluating researchers (e.g., professors, deans) have grown numb to the problems with the current publication system. As a consequence, they may fail to understand the reason, source, and gravity of untenured researchers' dissatisfaction with the publication system. Moreover, it suggests that those in power to change the publication system are the ones who are less likely to do so because they are the ones who are comfortable within the system as it currently is [13].

To go beyond the stage of merely identifying potential problems and instead proactively contribute towards a potential solution, the second aim of this paper is to conceptually envisage an alternative. As we consider prolonged publishing times to be a problem that is inherent to the journal-based publication system, we propose a radically different, holistic solution. For this, we build on the recently advanced idea of a digital "research-as-you-go" publication system [14] to depict how we can move beyond the journal-based publication system. Specifically, we propose a modular system [15] that enables researchers to communicate their contributions immediately when they are ready. Such

---

[1]　　We argue below that the journal impact factor (JIF) is a flawed measure of article quality. However, in our empirical context, the JIF still represents a reliable measure of journal desirability, because it is still widely embedded in academia as a relevant measure of quality. The JIF is therefore likely to influence researchers' satisfaction with a specific publication and the corresponding publication process.

a system thus eliminates publishing times altogether and has various other advantages that we will outline in the second, conceptual part of this paper.

As such, our partly empirical and partly conceptual paper provides three main contributions to the field. First, we contribute to the mounting evidence that researchers are dissatisfied with the current publication system. In focusing on publishing times, we build a theory and provide evidence regarding the reasons for researchers' dissatisfaction. Second, we address why, despite its problems and many criticisms, the current journal-based publication system is so persistent by arguing and showing that tenured researchers tend to be less dissatisfied with the current publication system compared to untenured researchers. Third, there has been very little reflection among Work and Organizational Psychology (WOP) researchers about cognitive alternatives for the current publication system [16]. Rather than proposing some incremental changes, we envision what an alternative publication system for (WOP) research could look like.

## 2. The Journal-Based Publication System: Paradise Lost?

Scholars have argued that a publication system should serve five functions [17,18]. The first is registration, which refers to the administration that a study has taken place and enables claiming precedence of a scholarly contribution. In the current journal-based publishing system, this occurs when a paper is published. Second, certification involves the process via which the contribution is validated. This typically occurs via peer review and editorial guidance. Third, awareness entails the ways in which scientific contributions are disseminated among stakeholders. In the last two decades there has been a shift from dissemination via printed journals to the online dissemination of articles, but most research is still submitted to journals, after which it is printed and published by their respective publishers. Fourth, archiving indicates that a publication system should keep a record over time of all scholarly contributions. Journals as well as repositories do this online, whereas libraries usually keep printed copies. Finally, the fifth function of a publication system is rewarding, which prescribes that the system should generate metrics that can be used for evaluating and rewarding researchers. Such metrics are predominantly journal-based (e.g., impact factor), but there is an increase in metrics at the individual researcher level (e.g., citations, h-index) as well as so-called altmetrics which are on the article level (e.g., the amount of reads or the amount of mentions an article receives in social media).

In the pre-digital era, the journal-based publishing system was a relatively effective and optimal way to fulfill each of these five functions. Limitations in addressing a function were generally due to external constraints (i.e., in time, space, and/or money). With the dawn of the digital era, many of these constraints have—in theory—fallen away; in some ways the publication system is changing accordingly, as is, for example, evidenced by the transition to the online dissemination and archiving of articles. However, in most other ways the publication system—in practice—has largely remained the same compared to the pre-digital age [19], which entails that many limitations are still in place.

Examples of such limitations regarding the registration function of the journal-based publication system include, for example, that the journal-based publishing system causes a bias in the scholarly contributions that are registered [20] and only registers studies after they have been conducted and accepted for publication—which can cause a delay of many years. To overcome publishing time, studies can be published before being reviewed (e.g., via preprint servers), but preprints are still not often used and generally not considered as official academic output in the field of WOP.

Whereas peer review is generally considered the best way for certification, the general limit of two-to-three reviewers and a single editor makes the current system prone to biases: which reviewers are selected; the extent to which the reviewers and editor favor the topic and approach of the paper; the mood and experience of the reviewers and editor; the extent to which the judgment of the reviewers and editor is affected by style and grammar issues; etc. [21]. A potential solution can be found in enabling everyone to write a review, which is very common for most products and services nowadays [22], but still underused in academia.

With the vast majority of papers stored behind paywalls, there is still a major limitation in the awareness and archiving functions of the current publication system that severely hampers discovery and innovation, and that exacerbates inequalities between those who have access and those who do not. Interestingly, even Harvard Library announced in 2012 that it could not afford to pay the subscription fees for all journals [23], which means that even one of the richest university libraries in the world cannot provide its researchers access to all scholarly works. The Open Science movement is a collaborative initiative to make scientific research and data accessible to all by removing paywalls [24], but to date is limited in its effects due to the vested interests and power of academic publishers as well as—what we argue in our paper—scholars who have grown numb to the problems with the current publication system and/or achieved a comfortable position in it.

Finally, concerning the rewarding function, the current journal-based publishing system predominantly evaluates and rewards researchers based on journal metrics such as impact factor. Not only is this based on the false assumption that journal impact factor predicts the impact of an individual paper [25]; it also entails that researchers are only rewarded after a paper has been accepted for publication in a journal, which is subject to the many biases mentioned above and may occur several years after the majority of the work (i.e., conceptualization, data collection, data analysis, drafting of the paper) has taken place. Whereas there have been ample calls to abandon the reliance on journal metrics in rewarding individual researchers [26], to date it remains common practice in most academic departments that conduct WOP research.

To summarize, there is an increasing awareness of the flaws of the current publication system [15], but the extent to which it is changing is limited at best. It is therefore not strange that there is a growing dissatisfaction among academics with the current journal-based publication system. Whereas there are many potential reasons for dissatisfaction, in the remainder of this paper we focus on publishing time for two reasons. First, publishing time is core to a number of the limitations mentioned above. Registration, awareness, and rewarding only happen when a paper is published, whereas the bulk of work on a paper may have occurred years earlier. This delay is likely to come across as unfair and unnecessary, which leads to increasing dissatisfaction. Moreover, although certification is a generally agreed-upon cause of publishing times taking one or multiple years, it is questionable whether this procedure of validation is perceived to justify the long publishing times. Second, because rewarding in academia predominantly occurs based on published papers (and the journals in which they are published), publishing time can have a pervasive impact on the success or failure of a researcher's career. As such, out of all limitations of the current journal-based publication system, we argue that publishing time has the most severe impact on a researcher's career and is therefore likely to have a strong bearing on researchers' dissatisfaction with the current publication system.

## 2.1. Publishing Time Dissatisfaction, Journal Desirability, and Reviewer Quality

The certification function of the journal-based publication system bestows publishing time with legitimacy: to ensure a study's validity and reliability, time is needed for external reviewers and editors to assess the quality of the work. From a distributive justice point of view, we argue that researchers whose paper is accepted for publication are therefore likely to approve of some amount of publishing time. The specific potential reward that can be the outcome of the certification function is that a study may be accepted for publication and hence receive the public recognition of adhering to certain qualitative standards.

To capture this distributive component of publishing time satisfaction, we focus on the extent to which researchers find the publishing time worthwhile. Worthwhile in this context refers to the extent to which attaining a publication is worth the process to achieve that outcome. A number of studies [10,27] suggests that a more desirable outcome reduces the extent to become dissatisfied with an unfair procedure. As such, whereas we expect that publishing time overall decreases the extent to which researchers perceive publishing time to be worthwhile, we argue that this relationship is

buffered by the perceived desirability of the journal in which their paper is published. Accordingly, we hypothesize:

**Hypothesis 1a (H1a):** *The desirability of the journal where a paper is published buffers the extent to which publishing time is negatively associated with the extent to which authors perceive publishing times to be worthwhile, such that the negative effect of publishing time on worthwhileness decreases when journal desirability increases.*

The extent to which employees are satisfied with a particular outcome does however not only depend on distributive justice, but also on procedural justice. Employees who receive equal outcomes can differ in their satisfaction when they (perceive they) have been treated differently [28]. Regarding publishing times, this suggests that researchers whose paper got accepted but had a longer publishing time compared to an accepted paper of other researchers (or another accepted paper by themselves) may feel they have been treated less fairly [29] and may therefore be less satisfied with the publishing time. From a procedural justice point of view, we therefore argue that longer publishing times are likely to negatively affect publishing time satisfaction.

To capture the procedural component of publishing time satisfaction, we focus on the extent to which researchers find the publishing time justifiable. The justifiability of publishing times refers to the extent to which researchers perceive there to be a good reason for publishing times to take as long as they do. Overall, we expect publishing time to negatively affect the extent to which researchers perceive publishing times to be justifiable, because a longer publishing time suggests that the procedure was prolonged whereas the outcome remained the same. However, if researchers perceive there to be a good reason for such a prolonged procedure, they may be more forgiving and hence become less dissatisfied with longer publishing times. As the certification function rests primarily in the hands of the reviewers, we argue that the perceived quality of the reviews is key here. The more that researchers perceive the reviewers to provide insightful reviews that improve the paper, the more likely it is that researchers consider the publishing time to be justifiable. Accordingly, we hypothesize[2]:

**Hypothesis 1b (H1b):** *Perceived review quality buffers the extent to which publishing time is negatively associated with the extent to which authors perceive publishing times to be justifiable, such that the negative effect of publishing time on justifiability decreases when perceived review quality increases.*

*2.2. Publishing Time Dissatisfaction and Tenure*

We argue that the extent to which researchers are dissatisfied with longer publishing times is also likely to depend on whether researchers are tenured or untenured. For untenured researchers, their precarious situation makes injustice more consequential to them than to tenured researchers [30]. Indeed, prolonged publishing times can cause untenured researchers to be denied tenure and pushed out of academia, whereas for tenured researchers they may cause a (temporal) delay in promotion or cut in research time at worst. Given that the stakes are thus much higher for untenured researchers, we expect that they will become more dissatisfied when the publishing time is prolonged and expect this to be true for the distributive as well as the procedural component of publishing time satisfaction (i.e., the extent to which the publishing time is worthwhile versus justifiable, respectively).

Furthermore, tenured researchers may become less dissatisfied when publishing times are longer because they are likely to have grown numb to prolonged publishing times and consider it a necessary evil. System justification theory suggests that individuals tend to legitimize the systems and social

---

[2] Hypothesis 1b is not reported in our preregistration of hypotheses because it was added based on an abductive process after seeing the strong direct effect of perceived reviewer quality on publishing time justifiability. The preregistration can be found here: https://osf.io/9zrxv/?view_only=228b20592b4142579147d030123a1647.

arrangements in which they live and work, even at the expense of personal and group interests [12,31,32]. Tenured researchers may thus consider prolonged publishing times as an imperfection that is inherent to an otherwise functional or even optimal system. Moreover, in having achieved tenure in the current system, they have achieved a comfortable, desired position, which means that imperfection is likely to have little negative effect to them and may even cause them to defend the system [33].

Based on justice as well as system justification theories, we therefore expect that researchers without tenure are more likely dissatisfied with prolonged publishing times compared to tenured researchers. We do not expect there to be a difference in the extent to which these relationships would be stronger or weaker for the worthwhileness or justification components of publishing time satisfaction, but provide two separate hypotheses in order to remain consistent with Hypotheses 1a and 1b:

**Hypothesis 2a (H2a):** *Hierarchical rank buffers the extent to which publishing time is negatively associated with the extent to which authors perceive publishing times to be worthwhile, such that the negative effect of publishing time on worthwhileness is lower for tenured compared to untenured researchers.*

**Hypothesis 2b (H2b):** *Hierarchical rank buffers the extent to which publishing time is negatively associated with the extent to which authors perceive publishing times to be justifiable, such that the negative effect of publishing time on justifiability is lower for tenured compared to untenured researchers.*

## 3. Methods

### 3.1. Research Strategy

Since we focus on the time between when a paper is submitted and accepted for publication, our study focuses on researchers who recently published their research. Whereas publishing time dissatisfaction is likely to be particularly high for studies that have been submitted but not yet published, an exclusive focus on published papers helps us to assess whether the ends (i.e., having a paper accepted for publication) are considered to justify the means (i.e., publishing time). This entails that our sample of researchers is likely to be positively biased in their satisfaction with the current publishing system. We argue that if we find support for our hypotheses in this sample, the negative effects of publishing time on researchers' satisfaction are as large or larger in the entire population of researchers, which includes unpublished researchers. Whereas studies usually focus on contexts where they expect to find support for their hypotheses, our study thus focuses on a context where support for our hypotheses is the least likely. This entails that we should be able to generalize our findings to a broader context if we find support for our hypotheses in this sample.

We opted for a cross-sectional survey to gather information on publishing time and the quality of the reviews and editorial guidance while preserving anonymity. Given that the main independent variable (publishing time) as well as two of the three moderating variables (i.e., journal desirability and hierarchical rank) focus on objective data, we deemed the risk for single source bias to be minimal. We preregistered the hypotheses, sample size, data collection procedures, and analytic techniques at: https://osf.io/7bdmh/?view_only=228b20592b4142579147d030123a1647.

### 3.2. Procedure, Sample, and Data Collection

We distributed the survey in two ways. First, we sent an online survey link to researchers who recently published in WOP journals. We focused on WOP journals to ensure comparability among journals, and because they tend to have long publishing times. We compiled a list of 15 prominent WOP journals[3] and collected contact information from the first author of every original research article

---

[3]    The journals used were Personnel Psychology, Journal of Applied Psychology, Journal of Organizational Behavior, Journal of Occupational Health Psychology, Organizational Behavior and Human Decision Processes, European Journal of Work and

in the four most recent issues at the time of data collection (November 2018). We did not sample authors from special issues, because the limited time frame for special issues could lead to selection bias on the independent variable. When the contact information of the first author was unavailable, we sent the survey to the corresponding author. We also distributed the online survey to 35 WOP scholars in our own network.

Respondents could fill out the survey multiple times if they had more than one first-authored publication. The 553 invitations we sent yielded a response of 194 (response rate: 35.08%), including 13 entries by respondents that filled out the survey more than once. We sent out three reminders to all authors in the sampling frame. After removing responses that contained missing data, we had 141 responses for publishing time justifiability and 145 responses for publishing times worthwhileness. Approximately 48.5% of the sample had tenure, while 53.5% of the respondents identified as female. Anonymized data, analyses, and supplementary files are all publicly available at: https://osf.io/b8pzf/?view_only=0019531da41b419dac625ad9e7abffbf.

### 3.3. Measures

The two dependent variables were publishing time justifiability and worthwhileness. Justifiability was measured with a single item: "To what extent do you consider the time it took between the date of original submission at the journal where it got accepted and final acceptance to be justifiable compared to the extent to which your manuscript has changed in the meantime?" We measured worthwhileness with the same item but replaced justifiable with worthwhile. Participants responded to the items on a 7-point Likert scale (from 1 "Strongly disagree" to 7 "Strongly agree").

The independent variable was operationalized as time to publish. Participants were asked to indicate the amount of years and months that passed between the submission to the last journal the article was submitted to and acceptance at that journal.

Regarding the moderators, journal desirability was operationalized by looking at the journal impact factor, given that publishing in top-tier journals tends to be valued more [34]. Journal impact factor was manually collected from the Journal Citation Reports of Clarivate Analytics, from which we used the two-year impact factor. Perceived review quality was measured on a single item asking respondents to rate the quality of the reviews at the journal where their article was accepted on a 7-point Likert scale which ranged from 1 "Extremely bad" to 7 "Extremely good". The survey contained a closed question on academic rank, which we used to measure those tenured. Participants were given the possibility to choose among: 'full professor', 'associate professor', 'assistant professor with tenure', 'assistant professor without tenure', 'post-doc', 'PhD student', 'teacher', or researcher'. We collapsed the first three categories into the dummy 'tenured', while the remaining categories were collapsed into the 'non-tenured' dummy.

We controlled for a number of variables. We controlled for the number of journals the manuscript was submitted to and whether the respondent perceived the manuscript to have improved between submission and acceptance to the journal in which it was published (a 7-point Likert scale was used which ranged from 1 "Much worse" to 7 "Much better"). We also controlled for the *quality of the editorial guidance* at the journal where their article got accepted (on a 7-point Likert scale which ranged from 1 "Extremely bad" to 7 "Extremely good"). Last, we included the respondent's gender (male as the reference category) and age.

---

Organizational Psychology, Organizational Psychology Review, Journal of Occupational and Organizational Psychology, Journal of Vocational Behavior, Human Resource Management, Human Resource Management Review, Leadership Quarterly, Journal of Business and Psychology, Work and Stress, and Human Factors.

## 4. Results

Table 1 presents the descriptive statistics and correlations for all variables. We found that respondents are on average satisfied with their publishing experiences (justifiability$_{mean}$ = 5.38; worthwhile$_{mean}$ = 5.57). While worthwhileness is on average the same for researchers with or without tenure (worthwhileness$_{tenure}$ = 5.53; worthwhileness$_{non-tenured}$ = 5.57), justifiability is on average lower for non-tenured researchers (justifiability$_{tenure}$ = 5.45; justifiability$_{non-tenured}$ = 5.26). We also found that the average time it takes authors to publish at the last journal that they submitted to is 1.19 years, while the longest amount of time to publish in our sample is 6 years. Last, we also found that the variables, justifiable and worthwhile, are highly correlated ($r$ = 0.93). We still performed separate analyses for the two dependent variables, as we expected different moderators to influence the effect of time to publish. Moreover, we found different results for justifiability and worthwhileness, further confirming that these are two distinct factors in publishing satisfaction.

To test the effects of time to publish on perceived justifiability and worth, ordinary least squares (OLS) regressions were conducted. The reported significance levels for the hypothesis tests are one-sided *p*-values as we used directional hypotheses. In line with our general argument and expectations, we found that publishing time is negatively related to worthwhileness ($\beta$ = −0.21, $p$ = 0.05) and justifiability ($\beta$ = −0.42, $p$ < 0.01).

Table 2 displays the models estimating perceived worth. Hypothesis 1a is not supported (Model 2), as journal desirability does not significantly moderate the relationship between time to publish and perceived worth ($\beta$ = −0.03, $p$ = 0.33). We failed to find support for Hypothesis 2a (Model 3), as tenure does not significantly moderate the relationship between time to publish and perceived worth ($\beta$ = 0.16, $p$ = 0.26).

Models in Table 3 examine the effect of publishing time on perceived justifiability. We found support for Hypothesis 1b (Model 2), as review quality has a significant moderating effect on the relationship between publishing time and perceived justifiability ($\beta$ = 0.41, $p$ < 0.01). This result implies that the negative effect of time to publish on perceived justifiability becomes less negative when review quality increases. In Figure 1, we display the dampening effect of review quality on the negative effects of time to publish.

We also found support for Hypothesis 2b (Model 3), as tenure has a significant effect on the relationship between time to publish and perceived justifiability ($\beta$ = 0.46, $p$ = 0.04). We plotted the marginal effects of the interaction between tenure and publishing time (Figure 2), which show that the negative association between time to publish and justifiability only decreases for untenured researchers. For researchers with tenure, time to publish does not affect their publishing time justifiability.

**Table 1.** Descriptive statistics.

| | M | SD | Min | Max | 1 | 2 | 3 | 4 | 5 | 6 | 7 | 8 | 9 | 10 | 11 |
|---|---|---|---|---|---|---|---|---|---|---|---|---|---|---|---|
| 1. Justifiable | 5.38 | 1.68 | 1.00 | 7.00 | | | | | | | | | | | |
| 2. Worthwhile | 5.57 | 1.49 | 1.00 | 7.00 | 0.93 | | | | | | | | | | |
| 3. Publishing Time | 1.23 | 0.83 | 0.08 | 6.00 | −0.28 | −0.23 | | | | | | | | | |
| 4. Tenure | 1.50 | 0.48 | 1.00 | 2.00 | 0.02 | −0.02 | 0.13 | | | | | | | | |
| 5. Journal Desirability | 3.14 | 1.39 | 0.77 | 8.86 | 0.07 | 0.05 | 0.15 | −0.13 | | | | | | | |
| 6. #Journals | 2.22 | 1.49 | 1.00 | 9.00 | 0.02 | −0.02 | 0.00 | 0.05 | −0.16 | | | | | | |
| 7. Improvement | 6.11 | 0.86 | 4.00 | 7.00 | 0.33 | 0.28 | 0.20 | 0.01 | 0.25 | −0.01 | | | | | |
| 8. Quality Reviews | 6.04 | 1.01 | 2.00 | 7.00 | 0.67 | 0.64 | −0.12 | 0.03 | 0.10 | 0.01 | 0.36 | | | | |
| 9. Quality Editorial | 6.17 | 1.23 | 2.00 | 7.00 | 0.59 | 0.60 | −0.21 | −0.02 | 0.20 | −0.02 | 0.18 | 0.66 | | | |
| 10. Gender | 0.43 | 0.54 | 0.00 | 1.00 | 0.05 | 0.06 | 0.02 | −0.06 | 0.04 | 0.06 | 0.03 | 0.17 | 0.14 | | |
| 11. Age | 37.99 | 8.50 | 25.00 | 80.00 | −0.04 | −0.07 | 0.00 | 0.22 | −0.09 | 0.02 | −0.1 | 0.07 | 0.10 | −0.11 | |

Note: $N = 154$–194 (please note that differences in $N$ are due to different numbers of missing values in different variables). Non-tenured was coded as 1, tenured was coded as 2. Female was coded as 1, male was coded 0. #journals represents the number of journals the manuscript was submitted to before it got accepted.

**Table 2.** Regression results—worthwhile.

| | Model 1 | | Model 2 | | Model 3 | |
|---|---|---|---|---|---|---|
| **Predictor** | **$b$** | **95% CI (LL, UL)** | **$b$** | **95% CI (LL, UL)** | **$b$** | **95% CI (LL, UL)** |
| (Intercept) | 0.18 | (−1.58, 1.94) | −0.04 | (−1.90, 1.83) | 0.50 | (−1.41, 2.41) |
| Publishing Time | −0.21† | (−0.45, 0.03) | −0.04 | (−0.56, 0.48) | −0.54 | (−1.35, 0.27) |
| Journal Desirability | −0.00 | (−0.37, 0.37) | 0.09 | (−0.17, 0.34) | −0.24 | (−0.90, 0.42) |
| Tenure | 0.01 | (−0.12, 0.14) | −0.02 | (−0.39, 0.35) | 0.02 | (−0.12, 0.15) |
| Final Journal Improvement | 0.14 | (−0.10, 0.39) | 0.14 | (−0.11, 0.38) | 0.16 | (−0.09, 0.40) |
| Quality Reviews | 0.64** | (0.40, 0.88) | 0.65** | (0.41, 0.88) | 0.64** | (0.41, 0.88) |
| Quality Editorial | 0.31** | (0.12, 0.50) | 0.31** | (0.12, 0.50) | 0.31** | (0.12, 0.49) |
| Age | −0.02* | (−0.05, −0.00) | −0.02* | (−0.05, −0.00) | −0.03* | (−0.05, −0.00) |
| Female | −0.18 | (−0.54, 0.19) | −0.19 | (−0.55, 0.18) | −0.18 | (−0.55, 0.18) |
| Final Time × Journal Desirability | | | −0.05 | (−0.17, 0.08) | | |
| Final Time × Tenure | | | | | 0.20 | (−0.26, 0.67) |

Note: $b$ represents unstandardized regression weights. LL and UL indicate the lower and upper limits of a confidence interval, respectively. † indicates $p < 0.10$. * indicates $p < 0.05$. ** indicates $p < 0.01$.

**Table 3.** Regression results—justifiability.

| | Model 1 | | Model 2 | | Model 3 | |
|---|---|---|---|---|---|---|
| **Predictor** | ***b*** | **95% CI (LL, UL)** | ***b*** | **95% CI (LL, UL)** | ***b*** | **95% CI (LL, UL)** |
| (Intercept) | −1.68† | (−3.63, 0.27) | 2.76* | (0.06, 5.45) | −0.90 | (−3.01, 1.20) |
| Publishing Time | −0.42** | (−0.69, −0.16) | −3.83** | (−5.37, −2.28) | −1.18** | (−2.05, −0.32) |
| Journal Desirability | 0.06 | (−0.08, 0.20) | 0.28 | (−0.09, 0.66) | 0.08 | (−0.06, 0.23) |
| Tenure | 0.20 | (−0.20, 0.60) | −0.06 | (−0.49, 0.36) | −0.35 | (−1.06, 0.36) |
| Final Journal Improvement | 0.22 | (−0.05, 0.49) | 0.09 | (−0.05, 0.22) | 0.24† | (−0.03, 0.51) |
| Quality Reviews | 0.69** | (0.42, 0.97) | 0.30* | (0.05, 0.56) | 0.69** | (0.42, 0.97) |
| Quality Editorial | 0.37** | (0.16, 0.59) | 0.31** | (0.10, 0.51) | 0.36** | (0.15, 0.58) |
| Age | −0.02 | (−0.04, 0.01) | −0.01 | (−0.04, 0.01) | −0.02 | (−0.04, 0.01) |
| Female | −0.22 | (−0.62, 0.18) | −0.26 | (−0.63, 0.11) | −0.24 | (−0.64, 0.15) |
| Final Time × Quality Reviews | | | 0.54** | (0.30, 0.78) | | |
| Final Time × Tenure | | | | | 0.46† | (−0.04, 0.96) |

Note: *b* represents unstandardized regression weights. LL and UL indicate the lower and upper limits of a confidence interval, respectively. † indicates $p < 0.10$. * indicates $p < 0.05$. ** indicates $p < 0.01$.

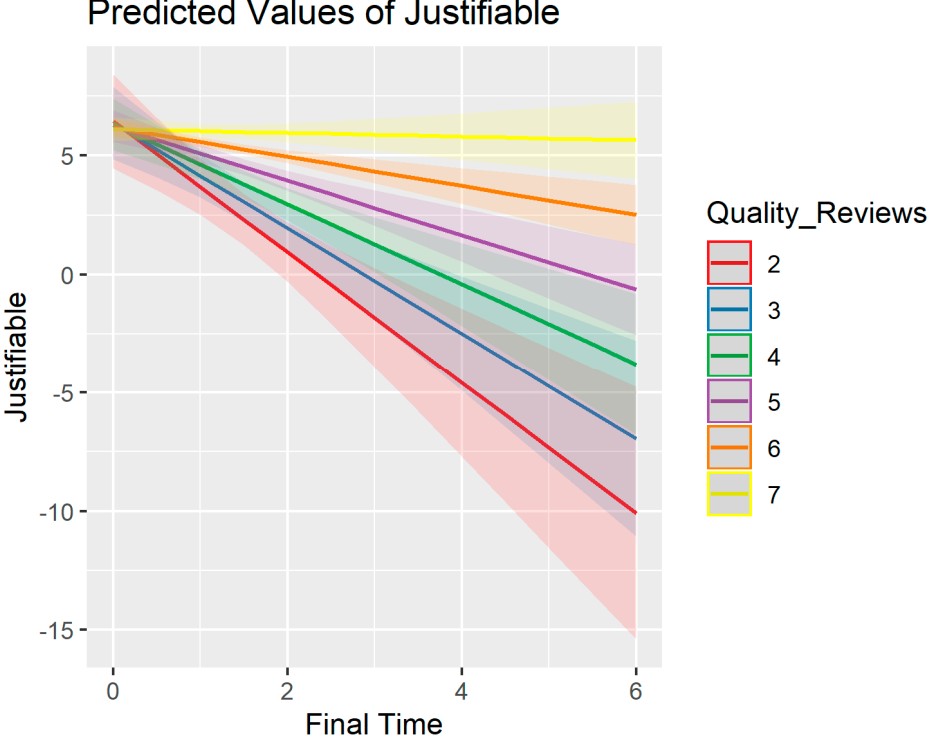

**Figure 1.** Review quality as a moderator of the relationship between publishing time and publishing time justifiability.

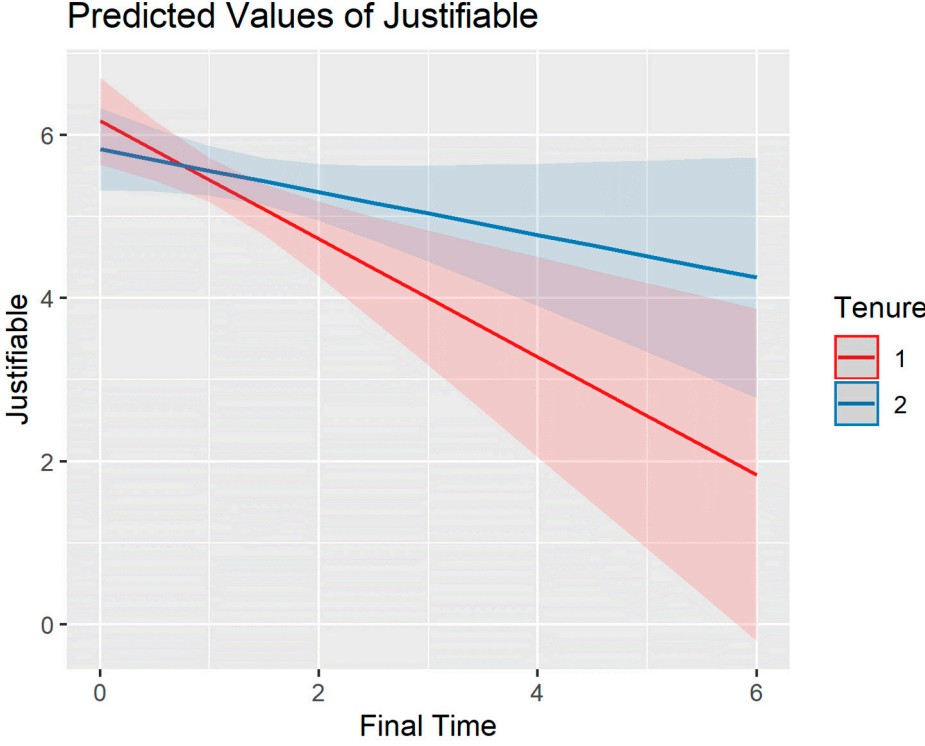

**Figure 2.** Tenure as a moderator of the relationship between publishing time and publishing time justifiability.

## 5. Discussion of the Empirical Study

At first blush, our findings seem to indicate that (WOP) researchers are overall satisfied with the current journal-based publishing system. The quality of reviews and the editorial guidance received high evaluations, and our respondents seem overall content with the publishing time of their paper. However, one must bear in mind that this sample represents the happy few (i.e., those that managed to get their paper published in a prominent (WOP) journal). Our sample thus is likely to be positively biased, which entails that the findings are likely to overestimate the satisfaction with the overall publishing system among the entire population of (WOP) researchers.

Furthermore, our findings indicate that there is a division between two categories of (WOP) researchers regarding how satisfied they are with the publishing time. Satisfaction with the publication system does not seem to change when publishing time increases for researchers with tenure, whereas among untenured researchers there is a sharp decrease in perceived justifiability the longer the publishing time lasts. In light of the likely positivity bias in our sample compared to the population of all (WOP) researchers, this is a worrisome finding for two reasons.

First, it suggests that researchers without tenure are aware of the perils of prolonged publishing times for their career. In showing that such prolonged publishing times are a primary source of dissatisfaction among untenured researchers, our study adds to the mounting pile of evidence showing that the current academic environment tends to be unhealthy for researchers in precarious jobs. For example, a recent study among graduate students (90% of which were PhD students) in 26 countries across 234 institutions showed that graduate students "are more than six times as likely to experience depression and anxiety as compared to the general population" [35]. Our study suggests that prolonged publishing times are likely to contribute to this "health crisis in academia" [16].

Whereas these findings warrant an improvement in the publication system, this is where the second reason to worry based on our findings kicks in. People tend to look to people in powerful positions to change the system. However, those in power (i.e., professors, editors, deans) generally hold tenured positions, and our findings indicate that their satisfaction with the current publication system is not affected by prolonged publishing times. As a consequence, they are less likely to experience a need to change the publication system, which means that change is unlikely to start with them [13]. This is not to say that tenured researchers are blind to the flaws of the journal-based publication system. However, they may have become comfortably numb or institutionalized by having found their way in the system and failing to see more feasible alternatives. As such, our study provides an answer to the question of why the journal-based publication system is still predominant in academia despite decades of calls for alternative publication systems inside and outside of academia [17] by showing that those with the power to instigate change are unlikely to do so.

An alternative, or additional explanation for the lack of improvements in our publication system may be a lack of cognitive alternatives [13]. Indeed, most studies that identified problems with different elements of the current publication system have focused on how those specific elements can be improved. As a consequence, they adopted a piecemeal approach to improve the current publication system [34,36]. In the following, we provide an overview of the most relevant improvements and discuss how adequately these improvements address the problems of publishing times that we have pointed out in this paper.

### 5.1. Current Solutions to the Publication System

To address design flaws in the current journal-based publication system, a number of solutions have been proposed and slowly implemented by funders, journals, as well as individual researchers. In line with the main theme of our paper, publishing time and researchers' satisfaction, we focus specifically on solutions that address publishing time and authors' control over their research output.

Preprints, the publication of a manuscript that has not yet been peer reviewed, has become more prominent as a method of communicating research. Preprints is a widely accepted method of communication in the natural sciences (e.g., ArXiv) and is becoming more prevalent in the social

sciences, but still hardly used in the WOP domain. Furthermore, while preprints are effective in increasing the speed of dissemination because it allows public communication of findings before the peer review process, it does not count as research output. For tenure or promotion, only papers that are published in peer-reviewed journals count. As such, whereas this practice addresses some issues with the journal-based publication system, it does not solve the problems regarding the publishing times.

Two related developments on timely dissemination that focus on reducing the negative impact of peer review are preregistration and registered reports. Preregistration reduces hypothesizing after results are known (HARKing) by separating confirmatory and exploratory hypotheses. Editorial processes and reviewers for journals in WOP often focus on changing the narrative to the data presented in a paper, which leads to potential hypothesis development after data has been collected. Next to the problems inherent to HARKing, this research practice tends to increase publishing time because it causes authors to focus on satisfying reviewers through recurring development of the narrative. Whereas preregistration thus may prevent HARKing and as such reduce publishing times, the reduction is likely to be limited because peer review will still take place prior to publication.

The registered report is a novel publishing format, where research proposals are peer reviewed prior to data collection [37]. If the registered report is accepted, the full paper will be accepted regardless of the findings. This format has been introduced to reduce the bias of mainly publishing papers with positive and novel results. This publishing format is highly potent in changing the way that we communicate research, as it redirects attention to important research questions instead of important results. Nonetheless, instead of reducing or eliminating publishing times, registered reports just move the review process forward. As such, the publication of a paper still depends on the subjective assessments of a few reviewers and one editor.

Taken together, we argue that preprints, preregistration, and registered reports are applaudable yet fragmented efforts to reduce the negative impact of questionable research practices as well as the negative impact of bias in the journal-based publication system. However, in terms of mitigating (the negative consequences of) publishing times, they are unlikely to make a big difference. Publishing times are inherent to the journal-based publication system. As a consequence, reducing publication times is unlikely to occur within this system, and certainly not via an incremental approach [38].

We propose that the only way to mitigate the negative consequences of publishing times is by providing reviews after, and not before, publication. Whereas such a practice is inconceivable within the journal-based publication system, it is in line with a recently proposed modular publication system [14]. This modular publication system was developed to offer a holistic alternative for the journal-based publication system, such that it not only focuses on improving the problems with publishing times, but also addresses several other problems of the journal-based publication system. In the following, we will briefly outline the modular publication system and its benefits, after which we will discuss how it can address the problems with publication times and advance (WOP) research in general.

### 5.2. An Alternative, Modular Publication System

Hartgerink and van Zelst (see [39], for a technical background) proposed a modular, digitally native publication system as an alternative for the current journal-based publication system. Their proposed publication system focuses on modules (i.e., pieces of research output that contribute to the current knowledge base). Modules can represent individual elements of the research cycle, for example a literature review, a hypothesis, or an analysis of a dataset. Modules can also consist of other contributions, including, but not limited to, a review of a module or an actual dataset. Since this modular publication system is digital, datasets can come in various forms and shapes, including audio and video recordings. In denoting the time when a module was published and what earlier modules it follows on, directed networks of modules are created. The emergent network provides insight into the knowledge base of the area of interest, and the synchronicity of the modules keeps track of the origin and provenance of each module and network. Via peer-to-peer protocols such a modular

publication system can be decentralized [14,39], which allows researchers to self-publish and retain ownership of their research while maintaining the 'open by design' principle that creates access for all.

This modular-based publication system holds the potential to outperform the journal-based publication system on each of the five functions of a publication system [17,18]. First, instead of registration being delayed by publishing times, it can take place the moment that a module is finished by publishing it online. As such, there will not be any prolonged publishing times, which means that the careers of researchers in precarious positions will no longer be at peril of biased reviewers/editors and delayed reviews or decision letters. Moreover, it inhibits the chances that researchers' ideas will be stolen by others and published before they can publish it themselves.

Second, rather than relying on the willingness, availability, and expertise of two or three reviewers who are prone to all sorts of biases, certification occurs by reviews of virtually anyone willing to provide a review. As such reviews also represent modules, they will be visible to everyone and can also be subject to review. Furthermore, a digital, modular system enables researchers with the opportunity to update their original modules, which could be based on reviews provided by others. As researchers will be more likely to build upon modules that are up-to-date and of high quality, there may be quite a strong incentive to do so. In contrast to the hypercompetition that is created by the current system, the modular publication system thus stimulates researchers to cooperate in building an up-to-date knowledge base.

Third, rather than being locked behind paywalls in the current journal-based publication system, the modular publication system is able to create awareness for anyone with access to a computer or smartphone. Because paywalls contribute to inequalities in terms of who can access, and therefore contribute, to scientific research, the modular publication system levels the playing field and paves the way for researchers from second- and third-world countries to read research and actively participate. Moreover, it enables practitioners at any location to access and capitalize on the most recent scientific insights and contribute where they can and desire to do so.

Fourth, instead of scholarly contributions being scattered among different journals and other contributions (e.g., reviews, datasets, null findings) being invisible, archiving occurs through a decentralized system of peer-to-peer protocols that are accessible to anyone. Moreover, archiving would no longer rely on one publisher as the responsibility would be decentralized to everyone who participates. In being digitally native, the modular publication system enables a more logical and appropriate way to store and retrieve scholarly contributions because they will be located in knowledge networks with other modules that focus on the same area of interest. Rather than having to search journals and online repositories for relevant papers, the modular publication system thus offers a much more convenient way of accessing the archive of studies on a particular topic when one identifies the relevant knowledge network.

Finally, regarding the rewarding function, the modular publication system offers a host of potential indicators that can be used for evaluating and rewarding researchers that are more proximal than the metrics of the journal-based publication system. Using a network approach, it would become possible to assess the centrality of individual researcher's modules in knowledge networks, the extent to which knowledge networks have originated from researchers' modules, and the extent to which the modules of individual researchers have been reviewed favorably. Such indicators provide much more valid measurements of the value of the contributions of individual researchers compared to the more distal and aggregated indicators like the impact factor of a journal where a paper of an individual researcher has been published. Moreover, because modules can also come in the form of scholarly contributions that in the journal-based publication system are generally not measured (e.g., software development, datasets, or the quantity and quality of reviews), the modular publication system also offers more comprehensive indicators that enable a more holistic evaluation of a researcher's contributions to science.

### 5.3. Implications for Publishing Times and (WOP) Research in General

The modular publication system abolishes publication times by giving researchers full ownership and letting them decide when a module is publishable. As such, delayed reviews cannot determine the careers of untenured researchers. Post-publication reviews may still be biased, but because anyone can provide a review, the bias of a single reviewer is less consequential. Furthermore, reviews represent only one of various elements used for evaluation and corresponding rewarding, which means that biases from reviewers will also have less impact on researchers' careers. We therefore argue that untenured researchers will be much more satisfied with a modular publication system compared to the journal-based publication system.

A more general problem with publication times is the delayed dissemination of scientific knowledge. By being able to immediately publish new discoveries, we argue that a modular publication system can greatly benefit the accumulation of knowledge. Moreover, a major advantage is the directed network structure of all modules. This is likely to create bodies of knowledge that center around a specific topic or area of interest and greatly enhances researchers' ability to identify and access relevant literature. For example, to address the question "What are the antecedents of proactive behavior at work?", in the current journal-based publication system, researchers need to search in online repositories for potentially relevant studies. Even after a thorough search, it is unlikely that all relevant studies have been identified (e.g., because of a mismatch of keywords, or because such studies are not listed in the repositories that were used). Identifying the relevant literature can therefore be a tedious and time-consuming job, which is why many researchers tend to rely on narrative and quantitative literature reviews. However, such reviews tend to suffer from the same problems. In contrast, the directed network structure of a modular publication system enables all studies that address a specific research question or hypothesis to be connected to that question or hypothesis. This entails that accessing the network pertaining to the question of interest will immediately provide one with an overview of all studies that have addressed that question. Furthermore, the modular system is open by design, which would allow for automated text mining and reading. In combination with improved possibilities for natural language processing, the efficiency with which scientific fields can be understood is greatly improved. We do not suggest that the directed networks render narrative or quantitative literature reviews obsolete, but they greatly facilitate making such literature reviews and, as a consequence, advance (WOP) researchers' ability to gain an overview of the state of the field of interest.

The directed network structure also facilitates the identification of the direction that research is going, as well as gaps in the literature. It could, for example, identify the extent to which literatures on similar topics have developed in separate networks. Identifying such separated knowledge networks that focus on the same or related topics would help researchers to know what gaps still exist in the knowledge base and what would be a good direction for future research to focus on.

Another major advantage of the proposed modular publication system for (WOP) research is that it facilitates publishing all findings, including null effects. The replication crisis in psychology has created awareness of the need to conduct replication studies [40,41] as well as of the file drawer problem [42]. However, there is a limited amount of places where such replications can be published, which means that conducting a replication may still represent a futile effort [43]. The proposed modular publication system may offer a solution, however. In connecting a module with null findings to modules containing the hypothesis that is addressed and related datasets and findings, such a module will be equally visible to researchers and hence can contribute as much to the knowledge base as positive or negative findings. Given that the file drawer problem is unlikely to be resolved in a hypercompetitive journal-based publication system that is biased towards quantitative, significant findings [44], we argue that such a change of the system—albeit radical—may be needed to overcome one of the main challenges facing (WOP) research nowadays. Indeed, the infinite space of a digital publication system enables publishing all scholarly contributions including null findings, which entails that publication bias and the file

drawer problem can be overcome. We refer the reader to Hartgerink [39] who explains in detail how a peer-to-peer system is able to realize 'infinite space' when enough peers adopt this alternative system.

Yet another major advantage of the proposed modular publication system for (WOP) research is its inclusiveness. (WOP) research shows how endemic exclusion is in all facets of organizational life [45,46], and academia is no better than any other sector [47,48]. To become more inclusive, we need to practice what we preach and remove barriers that prevent those who want to contribute [49]. This goes beyond taking down the paywall and includes being open to contributions in different styles and formats. A modular publication system would make that much easier because contributions would not need to fit the format of a journal and hence could consist of anything that can be made and shared digitally.

A related advantage to becoming more inclusive and open to different types of contributions is that a modular publication system can help reduce the science–practitioner gap because practitioners can more easily contribute (e.g., by providing testimonials, experiences, or datasets). In particular, in the management research community there have been calls for more evidence-based practices and interventions [50], and the best way to do that would be to make practitioners part of the scientific debates and discussions. Moreover, the proposed modular publication system would allow practitioners to provide specific research questions that researchers can address, thereby greatly reducing the science–practitioner gap that (WOP) research tends to struggle with.

### 5.4. Limitations

We acknowledge that our empirical study as well as modular publication system have several limitations that call for future research. Regarding our empirical study, the main limitation is that it was conducted in a single discipline (i.e., WOP), and our findings may partially reflect specific intricacies of this discipline. Among others, WOP is characterized by a relatively traditional focus on publishing in journals and a strong appreciation of higher ranked journals. In comparison to, for example, economics, there is no centrally located database of working papers, nor is there a place for discussing work in progress that is common in more technological disciplines. We call upon future research to address potential variations in disciplines, which may have consequences for the relationship between publishing times and researchers' satisfaction, and how that relationship differs between tenured and untenured researchers.

As for the modular publication system, we argue that it outperforms the journal-based publication system in all five functions of a publication system, but that is not to say it is without limitations or liabilities. We foresee three main potential liabilities with the modular publication system that need to be addressed adequately in order to bring it to fruition.

First, the utility of a new publication system is tied to its acceptance in the research community. If administrators (including tenure and promotion committees) and tenured faculty do not adopt the modular publication system, then contributions to it will not count for hiring, tenure, or promotion, thereby also limiting the incentives for untenured researchers to use it. As a consequence, it is paramount that administrators and tenured faculty embrace the modular publication system, yet this is where our findings suggest problems may arise. The more prominent and comfortable an institution or individual researcher is within the journal-based publication system, the lower the likelihood that they will endorse such a new system. We therefore consider promotion of the modular publication system to be crucial for its success. To convince administrators and tenured faculty to get on board of the modular publication system, we propose focusing on four arguments. The first is the content argument, by indicating how a modular publication system outperforms the journal-based publication system on each of its five functions. The second is the science argument, by pointing out that a modular publication system greatly advances the timely dissemination of research findings and the accumulation of scientific knowledge. The third is the reduced burden argument. In the journal-based publication system, tenured faculty spend a relatively long time on reviewing and editing. In a modular publication system, those tasks partly fall away, and partly are distributed more evenly among all researchers. Finally, the fourth

argument is the evaluation accuracy argument, as administrators currently rely on problematic proxies like quantity of publications and the JIF to evaluate researchers, but can rely on more accurate, varied, and personalized metrics to assess the quantity and quality of researchers' contributions within a modular publication system [14]. Notwithstanding this liability and recommendation, we however should also point out that not all administrators and tenured faculty are likely to be comfortably numb, as is evidenced by the fact that there are administrators and tenured faculty among the frontrunners of the push for changing the journal-based publication system. As a consequence, there may also be a special role for those early adaptors who may use their prominence and influence to help get others on board.

A second liability of a modular publication system is that granting researchers full ownership and allowing them to self-publish their modules runs the risk of massive volumes being dumped on the system, which may make it very difficult to navigate through content and discover relevant, high-quality contributions. In part, we consider this a liability that requires a technological solution in terms of organizing the data, optimizing search and selection algorithms, and detecting questionable content [39]. Lessons here can be drawn from open platforms like Linux and Wikipedia that face similar challenges [51]. Further, we posit that it will help to require researchers to create a profile, given that such a profile will allow researchers to build a track record and corresponding reputation. To avoid losing face, researchers will want to make sure that their modules are of high quality before publishing them (e.g., by asking for friendly reviews beforehand).

Third, quality may also be at stake by allowing anyone to review. This may lead to inflated ratings, or even researchers gaming the system and gathering many positive evaluations of their own contributions (e.g., by soliciting positive reviews, or creating fake profiles and using those for creating positive evaluations). We believe that the solutions proposed above may also help to mitigate these liabilities. Algorithms can help discover fake profiles that are only used to provide positive reviews of specific researchers' contributions (or negative reviews of rival researchers' contributions) as well as identify groups of researchers that provide each other with extra-ordinary positive ratings. Furthermore, because reviews also come in the form of modules that can be reviewed and form part of a researcher's profile, there is an incentive for researchers to only provide reviews of sufficient quality.

Whereas there are various other potential liabilities of a modular publication system that can be named, we argue that it overcomes the problems associated with publication times that we addressed in this paper, as well as various other issues with the journal-based publication system. As such, we consider it beyond the scope of this paper to further discuss the particularities, pros, and cons of a modular publication system here, and call for researchers and practitioners to continue the discussion and start working on its implementation.

## 6. Conclusions

Many researchers pursue an academic career because of an intrinsic motivation to contribute to a better world, but our study adds to a pile of evidence that the current journal-based publication system tends to inhibit researchers' motivation as well as the accumulation of scientific knowledge. In showing that untenured researchers are frustrated by prolonged publishing times whereas tenured researchers have grown numb to such design flaws, our findings suggest that those in power to change the system are less likely to do so. We hope that our proposal for a radically different, yet feasible, alternative publication system resuscitates researchers' intrinsic motivation and inspires tenured as well as untenured researchers to cooperate in realizing an improved publication system. There are more specific advantages of the proposed modular publication system for research that can be named, but we expect the ones we listed in this paper to be enough to show that it is time—and perhaps even overdue—for a radical change in the way that scientific research is published.

**Author Contributions:** Conceptualization, H.v.D. and M.v.Z.; methodology, M.v.Z.; formal analysis, M.v.Z.; data curation, M.v.Z.; writing—original draft preparation, H.v.D.; writing—review and editing, H.v.D. and M.v.Z. All authors have read and agreed to the published version of the manuscript.

**Funding:** This research received no external funding.

**Acknowledgments:** Previous versions of this article were presented at the 2019 European Association for Work and Organizational Psychology Congress in Turin and the 2019 Academy of Management Meeting in Boston. We would like to thank Chris Hartgerink and Matthijs Bal for their invaluable suggestions on an earlier version of this manuscript.

**Conflicts of Interest:** The authors declare no conflicts of interest.

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
