# Peer review of "Comfortably Numb? Researchers’ Satisfaction with the Publication System and a Proposal for Radical Change"

_publications, doi:10.3390/publications8010014_

Round 1

Reviewer 1 Report

This is an impressively research article, beginning with the authors' focus on publishing time and its correlation with faculty rank and researchers' dissatisfaction with the publication system. Also effective is the authors' use of the five intended functions of the publication system to demonstrate limitations of the current system and the potential the proposed modular system holds to address those limitations.

There are two areas that, in my opinion, need to be addressed. 

First, the assumption of "infinite space" in the digital realm (lines 533-534). A modular publication system such as the one proposed, particularly one that supports audio and video datasets (lines 431-432), would require a significant amount of server space, plus skilled web developers and systems administrators who can build and maintain such a system. The costs associated with this infrastructure and human labor demonstrate that the paywall that limits access to the current publication system is much harder to eliminate than this article assumes. 

In short, who is paying for this new system? Which administrators and/or tenured faculty are advocating for resources (financial and otherwise) to be allocated to build and maintain it? 

Secondly, why would administrators and tenured faculty advocate for this new system if they have become "comfortably numb" to the limitations of the current system? I believe that the answer to the question lies somewhere in the modular publishing system that the authors propose, but right now the answer just isn't there. Perhaps the new way of reviewing research would be attractive to tenured faculty, in that it expands the pool of reviewers, taking the burden off of a select few (often tenured faculty) who are called on to review this work. Plus it allows for incremental interventions throughout the process, rather than relying on reviewers to assess and evaluate finished research projects, which is quite a time-consuming task.

Another suggestion is to add some insight about how the modular system proposed would facilitate the process of tenure and promotion. The biggest liability, in my opinion, of adopting new/digital publication systems lies in tenure and promotion committees' unwillingness to accept or consider these new systems as valid outputs. It seems the paragraph addressing the rewarding function of the modular publication systems (lines 482-494) would be an excellent place to address this concern, as the authors are arguing that metrics of impact of a researcher's work would be more accurate in the new proposed system.

Author Response

This is an impressively research article, beginning with the authors' focus on publishing time and its
correlation with faculty rank and researchers' dissatisfaction with the publication system. Also effective is
the authors' use of the five intended functions of the publication system to demonstrate limitations of the
current system and the potential the proposed modular system holds to address those limitations.
Response: Thank you for your kind words.
There are two areas that, in my opinion, need to be addressed.
First, the assumption of "infinite space" in the digital realm (lines 533-534). A modular publication system
such as the one proposed, particularly one that supports audio and video datasets (lines 431-432), would
require a significant amount of server space, plus skilled web developers and systems administrators who
can build and maintain such a system. The costs associated with this infrastructure and human labor
demonstrate that the paywall that limits access to the current publication system is much harder to
eliminate than this article assumes.
In short, who is paying for this new system? Which administrators and/or tenured faculty are advocating
for resources (financial and otherwise) to be allocated to build and maintain it?
Response: We acknowledge the issue of bandwidth and storage space in a decentralized system of
research communication. However, if such a system would be adopted by enough peers and institutions,
the bandwidth and storage space could be sufficiently addressed. For institutions, we think especially of
universities and academic libraries, given that those would represent the impartial entities whose interests
align with a decentralized system of research communication. If they collectively agree to adopt such a
system, the costs and support of such a system for individual institutions don’t need to be very high –
especially not in comparison to the annual journal subscription fees.
We do however think that discussing this issue in the paper goes a bit beyond its scope. The other
reviewers already found our discussion of the modular publication system somewhat of a stretch, so for
how to practically organize such a system we refer you and the readers to Hartgerink (2019), who
discussed it in more detail. To that end, we added the following sentence after lines 533-534: “We refer
the reader to Hartgerink (2019), who explains in detail how a peer-to-peer system is able to realize
‘infinite space’ when enough peers adopt this alternative system.”
Secondly, why would administrators and tenured faculty advocate for this new system if they have
become "comfortably numb" to the limitations of the current system? I believe that the answer to the
question lies somewhere in the modular publishing system that the authors propose, but right now the
answer just isn't there. Perhaps the new way of reviewing research would be attractive to tenured faculty,
in that it expands the pool of reviewers, taking the burden off of a select few (often tenured faculty) who
are called on to review this work. Plus it allows for incremental interventions throughout the process,
rather than relying on reviewers to assess and evaluate finished research projects, which is quite a timeconsuming task.
Response: Thank you for raising this point. We agree that the adoption by administrators and tenured
faculty is paramount in launching the proposed system, but that this indeed may be a challenge when
they are comfortably numb. In our revised manuscript, we have expanded the limitations section to also
discuss some limitations of the modular publication system. We added this as the first point, and included
a number of suggestions regarding how administrators, tenured faculty, and tenure and promotion
committees (in response to your next point) may be persuaded to adopt the new system. In part in line
with your suggestions, these are (1) content arguments as outlined in how the new system addresses the
five functions of a publication system, (2) being on the frontier of new scientific developments, (3)
reducing the burden of reviewing and editing, and when doing so, receiving credits for it, and (4) providing
more personalized metrics on assessing the quantity and quality of researchers’ contributions. For the full
part on this, see lines XXX-YYY.
Another suggestion is to add some insight about how the modular system proposed would facilitate the
process of tenure and promotion. The biggest liability, in my opinion, of adopting new/digital publication
systems lies in tenure and promotion committees' unwillingness to accept or consider these new systems
as valid outputs. It seems the paragraph addressing the rewarding function of the modular publication
systems (lines 482-494) would be an excellent place to address this concern, as the authors are arguing
that metrics of impact of a researcher's work would be more accurate in the new proposed system.
Response: Thank you for this suggestion. We agree that this is indeed another potential liability and have
in part added it to the limitations, and also added a bit more information in the paragraph on the rewarding
function to explain how a modular publication system provides more personalized metrics that enable
more accurate assessments of the quantity and quality of researchers’ contributions. However, to remain
within the scope of the paper, we did not go too much into this topic. We refer the reader to Hartgerink
and van Zelst (2018), who have discussed the rewarding function of a modular publication system more
in-depth.
Thank you very much for your positive and constructive suggestions, they really helped us to further
improve our paper

Reviewer 2 Report

This is an interesting paper and I really appreciated that you had pre-registered it. 

My major comment is that this article reads as a paper of two halves - to the point where it may be more feasible to make it two papers. The first being the study, looking at satisfaction levels with a specific factor of the publication process (time) and the perception of this on different groups of researchers, which, on the whole, is interesting work, with appropriate discussion with awareness of the bias in the data collected.

The second part of the paper is an interesting proposal, that I agree with in many ways, but with the exception of the link with dissatisfaction from untenured researchers with the current process doesn't link with the first part. It would also benefit from a view of what already exists and is being developed in this area, and challenges regarding reward and recognition, detailed metadata, open peer review and so on.

I will address specific items below, but feel that either the link between Sections 1-5 and 6 needs to be more evidenced, or the paper submitted as two separate ones.

Introduction - beware of expanding conclusions from Work and Organisational Psychology into other disciplines indiscriminately, and the same across organisations.  e.g. line 41 removing 'that is predominant in most academic disciplines' keep within both the bounds of evidence and stops those not based in disciplines where journals dominate from disengaging. This also applies e.g. line 42 'mainly based of where and how much they publish' is different to 'Publication track record' which adapts to the nuance of the organisation/discipline/career stage.

Line 46 - It is arguable that this is more due of world rankings of research reinforcing the status quo - English language publications are cited more, so researchers publish in English because increasing citations is linked with University rankings. 

Line 49 - add references to the other flaws? 

Line 55 - remove 'ie' and change to 'defined as'? In some disciplines annual journal publishing may leave many months between acceptance and first public availability.

Line 62 - define DEF

Line 64 - impact factor is problematic here - line 109 and 154 address it further, but JIF is inherently flawed as a measure of article quality - but researchers perception of the value of impact factor is still widely embedded. Some acknowledgement of this tension would be welcome.

Line 110 - output/article level metrics should be mentioned here. While, sadly, far from universal, there is a growing increase in the use of responsible research indicators and a turning away from e.g. H-index which, with a discipline, roughly correlates to number of years in academia.

Line 124 - This needs some nuance - while this may hold true in this specific field, in others (e.g. economics) working papers/preprints are valued highly and are a key part of the disciplinary conversation.

Line 125-131 - the reviewers are also likely to be the published authors and thus satisfied with the system, reinforcing the status quo. Line150 - 'most' may not be appropriate - large sections of arts and humanities research have never used journal based metrics.

Line 267 - of these 15 journals, all are hybrid journals - no fully open or born digital journals (my rough count is Wiley & Elsevier 4 each, APA, T&F and Sage 2 each, Springer 1) - publication time is traditionally longer in these and some consideration should be given to the tenured researchers who have moved away from closed or hybrid open publication - those senior researchers dissatisfied with the current publication model have the option to publish outside of the traditional 'top' journals as they already have tenure.

Line 398 - should be 'blind'?

Line 410-412 - some consideration on the impact of open research could be given here (as a major change to the overarching publication system) Section 6: These are interesting ideas, but may sit better as an opinion or discussion piece - it would also benefit from a view of what already exists and is being developed in this area, and challenges regarding reward and recognition, detailed metadata, open peer review and so on. Much of what is proposed exists, but isn't linked - preregistration, open data, open methodology, informal communications (e.g. blogs), social media engagement, open peer review, systems such as publons, Kudos, slideshare, github, etc. do much of this work, but the need for integration (e.g. DOI, Orcid, OrgID's and associated metadata) is clear.

Generally, I think it is an interesting piece of work, although it would benefit from some contextualisation of the cutting edge changes in the Scholarly communication/open publishing landscape, such as responsible research indicators and open platforms.

Author Response

This is an interesting paper and I really appreciated that you had pre-registered it.
Response: Thank you for your kind words and appreciation of the preregistration!
My major comment is that this article reads as a paper of two halves - to the point where it may be more
feasible to make it two papers. The first being the study, looking at satisfaction levels with a specific factor
of the publication process (time) and the perception of this on different groups of researchers, which, on
the whole, is interesting work, with appropriate discussion with awareness of the bias in the data
collected. The second part of the paper is an interesting proposal, that I agree with in many ways, but with
the exception of the link with dissatisfaction from untenured researchers with the current process doesn't
link with the first part. It would also benefit from a view of what already exists and is being developed in
this area, and challenges regarding reward and recognition, detailed metadata, open peer review and so
on.
I will address specific items below, but feel that either the link between Sections 1-5 and 6 needs to be
more evidenced, or the paper submitted as two separate ones.
Response: Thank you for your appreciation of the value of our work and the discussion we put forward.
We understand your point that the article reads as a paper of two halves, and have discussed among
ourselves whether we think it is better to split it into two papers, or not. We have decided not to do that
because of three reasons. First, we still believe that the issues we identify in our empirical part warrant
radical change given that publication times (and different perceptions of it depending on researchers’
tenure) are inherent to the journal-based publication system, so we don’t see a feasible incremental
solution to this problem. In proposing such a radical alternative, we need to explain it in full and can’t just
only focus on the element of the modular publication system that deals with how it eliminates publication
times. Second, Reviewer 1 showed particular interest in the proposed radical solution and asked
questions about elements of it that do not just relate to the issues outlined in our empirical part, which to
us signaled that there is indeed merit in keeping the two parts. Third, we fully agree with your point that
we should incorporate an overview of what already exists and is being developed, and realized that doing
so may provide a clearer bridge between the two parts. Upon re-reading our initial submission after
reading the reviews, we noticed that the connection between the two parts wasn’t as fluent as we thought
it to be. By adding a section in our revised manuscript on what developments exist and are happening in
the journal-based publication system, we believe we have created a clearer and more convincing
argument regarding why a radical alternative is warranted, and as such create more coherence between
the two parts. Specifically, in this part, we argue that previous proposals on improving research
communication have not done enough to tackle the central issue of our paper: addressing the pace of
information flow between researchers and the problematic consequences for researchers as well as
society. We refer the reviewer to part 5.1 of our paper for a detailed discussion on existing work and an
integration of both parts of our paper. We hope that you agree that in this way the article benefits from
having both parts in one paper.
Introduction - beware of expanding conclusions from Work and Organisational Psychology into other
disciplines indiscriminately, and the same across organisations. e.g. line 41 removing 'that is
predominant in most academic disciplines' keep within both the bounds of evidence and stops those not
based in disciplines where journals dominate from disengaging. This also applies e.g. line 42 'mainly
based of where and how much they publish' is different to 'Publication track record' which adapts to the
nuance of the organisation/discipline/career stage.
Response: Thank you for raising these issues. We agree that we can more nuanced here. We followed
your suggestions and indicated in the limitations that our findings pertain to the field of WOP, which is
characterized by holding on to the traditional journal-based publishing model, and that in other academic
disciplines the findings thus may be somewhat different.
Line 46 - It is arguable that this is more due of world rankings of research reinforcing the status quo -
English language publications are cited more, so researchers publish in English because increasing
citations is linked with University rankings.
Response: Thank you for this suggestion. We have added it as a potential cause of this phenomenon.
Line 49 - add references to the other flaws?
Response: We added a reference to a paper that addresses the mentioned flaws.
Van Witteloostuijn, A. (2016). What happened to Popperian falsification? Publishing neutral and negative
findings. Moving away from biased publication practices. Cross Cultural & Strategic Management, 23(3),
481-508.
Line 55 - remove 'ie' and change to 'defined as'? In some disciplines annual journal publishing may leave
many months between acceptance and first public availability.
Response: We agree with your suggestion here and replace ‘i.e.,’ by ‘defined as’. The variable we use
here is also measured by the time between submission and acceptance, not as time between acceptance
and availability to the reader.
Line 62 - define DEF
Response: We apologize as that should not have been in the paper. It indicated that the definition of
those constructs should be moved from later on in the paper to there, but we apparently did not do that
anymore. We have now done so.
Line 64 - impact factor is problematic here - line 109 and 154 address it further, but JIF is inherently
flawed as a measure of article quality - but researchers perception of the value of impact factor is still
widely embedded. Some acknowledgement of this tension would be welcome.
Response: Thank you for pointing out the tension here. We added a footnote at line 64 that addresses the
fact that we agree that the JIF is flawed, but use it in our empirical context exactly because it is still widely
used to assess the quality of researchers’ publications and therefore likely to influence our respondents’
satisfaction with a specific publication and the corresponding publication process. The footnote reads as
follows:
“We argue below that the journal impact factor (JIF) is a flawed measure of article quality.
However, in our empirical context, the JIF still represents a reliable measure of journal
desirability, because it is still widely embedded in academia as a relevant measure of quality. The
JIF is therefore likely to influence researchers’ satisfaction with a specific publication and the
corresponding publication process.”
Line 110 - output/article level metrics should be mentioned here. While, sadly, far from universal, there is
a growing increase in the use of responsible research indicators and a turning away from e.g. H-index
which, with a discipline, roughly correlates to number of years in academia.
Response: Thank you for addressing alternative metrics in your review. We agree and added the
following sentence after line 110: “as well as so-called altmetrics which are on the article level (e.g., the
amount of reads or the amount of mentions an article receives in social media)” (Sud & Thelwall, 2013).
Line 124 - This needs some nuance - while this may hold true in this specific field, in others (e.g.
economics) working papers/preprints are valued highly and are a key part of the disciplinary conversation.
Response: We appreciate the nuance you put forward here and addressed this by adding “in the field of
WOP” to line 124. The sentence now reads as: “but preprints are still little in use and are generally not
considered as official academic output in the field of WOP.”
Line 125-131 - the reviewers are also likely to be the published authors and thus satisfied with the
system, reinforcing the status quo.
Response: We fully agree - this is one of the major points of our empirical findings. But we think we make
this point throughout the paper and do not need to emphasize it also here.
Line150 - 'most' may not be appropriate - large sections of arts and humanities research have never used
journal based metrics.
Response: in line with the text we added in line 124, we also added “that conduct WOP research” after
line 150. The sentence now reads as: “to date it remains common practice in most academic departments
that conduct WOP research.”
Line 267 - of these 15 journals, all are hybrid journals - no fully open or born digital journals (my rough
count is Wiley & Elsevier 4 each, APA, T&F and Sage 2 each, Springer 1) - publication time is traditionally
longer in these and some consideration should be given to the tenured researchers who have moved
away from closed or hybrid open publication - those senior researchers dissatisfied with the current
publication model have the option to publish outside of the traditional 'top' journals as they already have
tenure.
Response: We appreciate the point you raise about fully open or born digital journals. With regards to the
sampling frame, most WOP researchers still mainly publish in the traditional hybrid journals because, to
our knowledge, there are no dedicated fully open journals in organizational psychology. While some
senior researchers sometimes publish in fully open journals, most continue to publish in traditional
journals because their promotion to higher ranks depends on it and they want to support their junior coauthors (e.g., in the context of papers born from dissertation manuscripts). We therefore believe that our
findings are generalizable to the wider context of WOP researchers. However, we do agree that there are
both within WOP as well as outside WOP tenured researchers who do want to change the system, so we
do not want to suggest that the tendency we observe among tenured researchers to be comfortably numb
applies to all tenured researchers. In our discussion, based on Reviewer 1’s suggestion, we already
discuss how tenured researchers can be persuaded to adopt the modular publication system. There we
now also acknowledge that some tenured researchers are actually frontrunners in moving beyond the
journal-based publication system.
Line 398 - should be 'blind'?
Response: Yes, thank you. ‘Blind’ is indeed the correct word here and we have adjusted this as such.
Line 410-412 - some consideration on the impact of open research could be given here (as a major
change to the overarching publication system)
Response: We agree that the paper would benefit from a discussion of what already exists and is
currently being developed in terms of open research, and as indicated above have incorporated a section
on that.
Section 6: These are interesting ideas, but may sit better as an opinion or discussion piece - it would also
benefit from a view of what already exists and is being developed in this area, and challenges regarding
reward and recognition, detailed metadata, open peer review and so on. Much of what is proposed exists,
but isn't linked - preregistration, open data, open methodology, informal communications (e.g. blogs),
social media engagement, open peer review, systems such as publons, Kudos, slideshare, github, etc. do
much of this work, but the need for integration (e.g. DOI, Orcid, OrgID's and associated metadata) is
clear.
Response: Thank you for pointing this out. As mentioned above, your suggestions helped us to improve
the connection between the two parts. In our revised manuscript, we have more clearly explained why
many of the valuable (!) earlier proposals to change research communication have adopted a piecemeal
approach, which leads either to incremental changes or to new disequilibria (in line with systems theory
thinking). We hope that you agree that our paper now better integrates the empirical and conceptual part
by providing a discussion of the progress that has already been made in this area and showing how the
modular publication system goes beyond that.
Generally, I think it is an interesting piece of work, although it would benefit from some contextualisation
of the cutting edge changes in the Scholarly communication/open publishing landscape, such as
responsible research indicators and open platforms.
Response: Thank you once again for your thorough and helpful comments and suggestions. They were
instrumental to our improved manuscript.

Reviewer 3 Report

This article has two basic sections. The first presents the results of a survey into researchers’ (dis)satisfaction with the present system of journal publication and in particular with peer review. The second section presents a suggested replacement for the journal publication system. The paper is well- researched, the argumentation is clear, and the subject is of interest.

The scope of the survey is limited since it involves a single sub-discipline (Work and Organizational Psychology). While this is justified in the context of a single piece of research one would hope that future research would look at the situation in other disciplinary areas such as the hard sciences, or the humanities, where the results might turn out to be different. I note that publishing time is defined as the time between paper submission and acceptance; however the time between acceptance and actual publication can also be fairly lengthy. The response rate of 35% seems fairly standard, but this still means that we do not know the attitudes of two-thirds of those to whom the questionnaire was sent.

The suggested alternative is put forward in the second part of the paper, and while the authors are fairly expansive on its advantages, they are less forthcoming on its limitations. They also say nothing about how the suggested alternative relates to recent developments, such as on-line ahead-of-print publication, open access publication, or the availability of supplementary materials on-line. I think it would be useful to say something about the fact that, if implemented, I would expect this system to produce a vastly increase volume of material, in a situation where researchers already find great difficulty in keeping abreast of the volume of research produced by the journal publication system. There also seems to be virtually nothing in the way of quality control, other than the fact that anybody can review anything. How could the researcher distinguish between valuable and less valuable research in this mass of material?

The journal publication/peer-review system is certainly not perfect, and while I would be sceptical about the feasibility of the system proposed, I think the debate on the question is useful and valuable, and this paper is a worthy contribution to that debate.

The paper is in general well written. I would, however, suggest the following as possible minor adjustments:

l.62 DEF – give the meaning of this acronym.

l.78 envision an alternative > envisage an alternative

l.155 because of two reasons > for two reasons

l.190 perceive to > perceive they

l.192 of themselves > by themselves

         may feel like they have been treated > may feel they have been treated

l.371 (WOP)researchers > (WOP) researchers [space after closed bracket]

l.403 those in power > those with power

l.434 in the knowledge > into the knowledge

435 & l.447 I don’t know what the authors mean by “according”. Perhaps “related”, but I think this word could be deleted.

l.555 are enough > to be enough

Overall, I found this an interesting paper which provides a useful contribution to an important debate. I recommend its publication with some minor revisions based on the suggestions in this report.

Author Response

This article has two basic sections. The first presents the results of a survey into researchers’
(dis)satisfaction with the present system of journal publication and in particular with peer review. The
second section presents a suggested replacement for the journal publication system. The paper is wellresearched, the argumentation is clear, and the subject is of interest.
Response: We are happy to read that you are so positive about our paper!
The scope of the survey is limited since it involves a single sub-discipline (Work and Organizational
Psychology). While this is justified in the context of a single piece of research one would hope that future
research would look at the situation in other disciplinary areas such as the hard sciences, or the
humanities, where the results might turn out to be different.
Response: Thank you for this suggestion. We added an explicit suggestion in the newly added limitations
section that future research should focus on the robustness of our findings across different disciplines,
given that there indeed is a large variation between disciplines in terms of publishing cultures.
Specifically, we added the following on this matter: “One limitation of our study is that it was conducted in
a single discipline (i.e., WOP), and our findings may partially reflect specific intricacies of this discipline.
Among others, WOP is characterized by a relatively traditional focus on publishing in journals and a
strong appreciation of higher ranked journals. In comparison to for examine economics, there is no
centrally located database of working papers, nor is there a place for discussing work in progress that is
common in more technological disciplines. We call upon future research to address potential variations in
disciplines, which may have consequences for the specific relationship between publishing times,
researchers’ satisfaction, and differences between tenured and untenured researchers.”
I note that publishing time is defined as the time between paper submission and acceptance; however the
time between acceptance and actual publication can also be fairly lengthy.
We fully agree and actually also collected data on that issue by asking respondents to indicate the time
between acceptance and first public availability. On average, this was about six months, while some
respondents had to wait for almost two years. We did not incorporate that into our analyses because most
journals publish papers online in advance, and already count for tenure and promotion decisions when
they are accepted. However, we agree that this issue is likely to exacerbate the more general problems
that we describe regarding the journal-based publication system. The alternative proposal that we put
forward also addresses this issue by giving the author control of when a module is published, and even
enabling post-hoc formatting and editing the publication.
The response rate of 35% seems fairly standard, but this still means that we do not know the attitudes of
two-thirds of those to whom the questionnaire was sent.
Response: The response rate of 35% is the normal standard in work and organizational psychology. As
we do not know characteristics of the respondents (they are measured in the survey), it was also not
possible to conduct nonresponse bias analyses.
The suggested alternative is put forward in the second part of the paper, and while the authors are fairly
expansive on its advantages, they are less forthcoming on its limitations. They also say nothing about
how the suggested alternative relates to recent developments, such as on-line ahead-of-print publication,
open access publication, or the availability of supplementary materials on-line. I think it would be useful to
say something about the fact that, if implemented, I would expect this system to produce a vastly increase
volume of material, in a situation where researchers already find great difficulty in keeping abreast of the
volume of research produced by the journal publication system. There also seems to be virtually nothing
in the way of quality control, other than the fact that anybody can review anything. How could the
researcher distinguish between valuable and less valuable research in this mass of material? The journal
publication/peer-review system is certainly not perfect, and while I would be sceptical about the feasibility
of the system proposed, I think the debate on the question is useful and valuable, and this paper is a
worthy contribution to that debate.
Response: Thank you for your appreciation of our contribution to an important debate and offering this
suggestion. Because of the scope of the paper, we were hesitant to dive too deep into the alternative
publication system, but agree that indicating some limitations/liabilities of the modular publication system
would be good to mention. In our revised manuscript, we highlight three liabilities. The first is that
implementing the new system and persuading researchers to use it may be difficult (in line with a
suggestion from Reviewer 1), and following your points, the second is that the system may contain
massive volumes of information that may make it very difficult to navigate, and third that allowing anyone
to provide a review may also vastly increase the number of biased reviews. For a full overview of these
points and some brief ideas regarding how those may be tackled, see lines 631-702 of the revised
manuscript.
The paper is in general well written. I would, however, suggest the following as possible minor
adjustments:
l.62 DEF – give the meaning of this acronym.
We apologize as that should not have been in the paper. It indicated that the definition of those constructs
should be moved from later on in the paper to there, but we apparently did not do that anymore. We have
now done so.
l.78 envision an alternative > envisage an alternative
l.155 because of two reasons > for two reasons
l.190 perceive to > perceive they
l.192 of themselves > by themselves
may feel like they have been treated > may feel they have been treated
l.371 (WOP)researchers > (WOP) researchers [space after closed bracket]
l.403 those in power > those with power
l.434 in the knowledge > into the knowledge
435 & l.447 I don’t know what the authors mean by “according”. Perhaps “related”, but I think this word
could be deleted.
l.555 are enough > to be enough
Response: Thank you for these suggestions. We implemented the suggestions unilaterally.
Overall, I found this an interesting paper which provides a useful contribution to an important debate. I
recommend its publication with some minor revisions based on the suggestions in this report.
Response: Thank you again for your kind words and for your thoughtful suggestions. They really helped
us to further improve our manuscript..

Round 2

Reviewer 1 Report

Thank you for your addressing my questions regarding questions of infinite space, labor, and cost of digital publishing and the modular system you propose. The edits made to the essay sufficiently address the concerns I had, and I think this is an excellent contribution to the ongoing conversation about much-needed changes in academic publishing.

Author Response

Dear reviewer,

Thank you for your kind words about our manuscript and for appreciating our revisions in response to your feedback. Thank you as well for constructively helping us develop our manuscript and we look forward to the discussion it hopefully inspires.

Kind regards,

Hans and Marino

Reviewer 2 Report

Firstly, I would like to thank the authors - first for both their kind words, and secondly for their obviously in depth, thoughtful, considered and comprehensive response to the feedback they received, both from me and the other reviewers. I welcome particularly their comprehensive engagement with the question of articulating the link between the two parts of the paper.

The redrafted paper has both nuance and clarity over the areas discussed, clearly articulating the limits of conclusions drawn from one area of study (WOP).

The information contained in section 5.1 and 5.4 considerably enhances the link between the study and the proposal, aligning the study and the proposals clearly and articulating the challenges and limitations.  This is a substantial addition to the paper and I’d again like to thank the authors for their consideration and adoption of the feedback on this point, the narrative through the paper is considerably strengthened.

One very minor comment – line 414 ‘proposed and are slowly implemented’ changes tense mid sentence, but has no impact on the quality of the paper.

I recommend that this paper is accepted in its present form & pass on my congratulations to the authors for an interesting and comprehensive article.

Sarah

Author Response

Dear Sarah,

Thank you for your kind words about our manuscript and our response to your constructive feedback. 

We have implemented your comment regarding line 414 in this revision, which now reads as: "a number of solutions have been proposed and slowly implemented by funders".

Thank you for constructively helping us develop our manuscript and we look forward to the discussion it hopefully inspires.

Kind regards, 

Hans and Marino

Reviewer 3 Report

This paper provides an interesting discussion of authors’ attitudes to publishing times in the current journal publication system. In this revised version the author has made a great effort to address the concerns of this reviewer. The subject is of interest and importance and merits the attention that this article recommends and contributes to. While I remain dubious about the feasibility of the proposed alternative system, it is nevertheless a useful contribution to the ongoing debate. I recommend publication of this paper.

I noted the following probably typos:

l.161 “decreasing dissatisfaction”: shouldn’t this be “increasing dissatisfaction”?

l.286 “mean time”: should be single word “meantime”.

l.577-8 “to contribute” this phrase is repeated twice.

Author Response

Dear reviewer,

Thank you for your kind words about our manuscript and for appreciating our revisions in response to your feedback. 

We have revised the manuscript in line with the suggestions you have proposed in this review.

Thank you as well for constructively helping us develop our manuscript and we look forward to the discussion it hopefully inspires.

Kind regards,

Hans and Marino